# Development of a Novel Chikungunya Virus-Like Replicon Particle for Rapid Quantification and Screening of Neutralizing Antibodies and Antivirals

Hui-Chung Lin,[a,b] Der-Jiang Chiao,[b] Pei-Yun Shu,[d] Hui-Tsu Lin,[b] Chia-Chu Hsiung,[b] Chang-Chi Lin,[b,c] (ID) Szu-Cheng Kuo[b,c]

[a]Graduate Institute of Medical Sciences, National Defense Medical Center, Taipei, Taiwan
[b]Institute of Preventive Medicine, National Defense Medical Center, Taipei, Taiwan
[c]Department and Graduate Institute of Microbiology and Immunology, National Defense Medical Center, Taipei, Taiwan
[d]Center for Diagnostics and Vaccine Development, Centers for Disease Control, Ministry of Health and Welfare, Taipei, Taiwan

**ABSTRACT** Chikungunya fever is a mosquito-transmitted infectious disease that induces rash, myalgia, and persistent incapacitating arthralgia. At present, no vaccines or antiviral therapies specific to Chikungunya virus (CHIKV) infection have been approved, and research is currently restricted to biosafety level 3 containment. CHIKV-like replicon particles (VRPs) are single-cycle infectious particles containing viral structure proteins, as well as a defective genome to provide a safe surrogate for living CHIKV to facilitate the testing of vaccines and antivirals. However, inefficient RNA transfection and the potential emergence of the competent virus through recombination in mammalian cells limit VRP usability. This study describes a transfection-free system for the safe packaging of CHIK VRP with all necessary components via transduction of mosquito cell lines using a single baculovirus vector. We observed the release of substantial quantities of mosquito cell-derived CHIK VRP (mos-CHIK VRP) from baculovirus-transduced mosquito cell lines. The VRPs were shown to recapitulate viral replication and subgenomic dual reporter expression (enhanced green fluorescent protein [eGFP] and luciferase) in infected host cells. Interestingly, the rapid expression kinetics of the VRP-expressing luciferase reporter (6 h) makes it possible to use mos-CHIK VRPs for the rapid quantification of VRP infection. Treatment with antivirals (suramin or 6-azauridine) or neutralizing antibodies (monoclonal antibodies [MAbs] or patient sera) was shown to inhibit mos-CHIK VRP infection in a dose-dependent manner. Ease of manufacture, safety, scalability, and high throughput make mos-CHIK VRPs a highly valuable vehicle for the study of CHIKV biology, the detection of neutralizing (NT) antibody activity, and the screening of antivirals against CHIKV.

**IMPORTANCE** This study proposes a transfection-free system that enables the safe packaging of CHIK VRPs with all necessary components via baculovirus transduction. Those mosquito cell-derived CHIK VRP (mos-CHIK VRPs) were shown to recapitulate viral replication and subgenomic dual reporter (enhanced green fluorescent protein [eGFP] and luciferase) expression in infected host cells. Rapid expression kinetics of the VRP-expressing luciferase reporter (within hours) opens the door to using mos-CHIK VRPs for the rapid quantification of neutralizing antibody and antiviral activity against CHIKV. To the best of our knowledge, this is the first study to report a mosquito cell-derived alphavirus VRP system. Note that this system could also be applied to other arboviruses to model the earliest event in arboviral infection in vertebrates.

**KEYWORDS** baculovirus, Chikungunya virus, mosquito cell, virus replicon particle

Chikungunya virus (CHIKV) is a mosquito-transmitted arbovirus that causes in humans a complex musculoskeletal inflammatory disease characterized by acute fever, rash, myalgia, and persistent arthralgia (1, 2). Prior to 2004, the primary vector

Address correspondence to Szu-Cheng Kuo, szucheng1234@gmail.com.

The authors declare no conflict of interest.

species of CHIKV was *Aedes aegypti*; however, viral glycoprotein mutants of CHIKV have expanded its vector competency to *Aedes albopictus*, which has led to global distribution (3, 4). There are currently no approved vaccines or antivirals to counteract CHIKV infection.

Chikungunya virus is classified as the genus *Alphavirus* in the family *Togaviridae*. The CHIKV virion is a spherical particle measuring ~70 nm in diameter comprising an icosahedral capsid with a positive single-stranded RNA genome (~12 kb) enclosed within a lipid bilayer embedded with E1, E2, and/or E3 glycoproteins (5–7).

The genome of CHIKV comprises a 5′ cap untranslated region (UTR), followed by two open reading frames encoding four nonstructural proteins (nsP1, nsP2, nsP3, and nsP4), and five structural proteins (C, E3, E2, 6K, and E1), as well as a 3′ UTR terminated by a poly-A tail (8). The polyproteins of nsP1 to nsP4 comprise the viral replication machine (6). The production of genomic and subgenomic (SG) RNAs is regulated by the processing of nonstructural polyproteins to form intermediates and eventually individual nsPs (9). The alphavirus replicase presents high activity in *trans* with the ability to replicate RNAs containing UTRs at both the 5′ and 3′ ends (10–13). In cases in which these RNAs contain SG promoter sequences, the SG RNA is also transcribed. These properties have been exploited in the packaging of alphavirus-like replicon particles (VRPs) (14, 15). SG mRNA (26S RNA), which is quickly transcribed from the SG promoter upon virus entry into the host cell (accumulating to $10^6$ molecules/cell), is involved in the translation of structural proteins for virion assembly.

The current classification of CHIKV as a biosafety level 3 (BSL-3) biological agent limits the use of infectious CHIKV in research and clinical laboratories. Research on CHIKV under lower-level safety containment is contingent on the development of a safe surrogate (16–19). The fact that alphavirus VRPs recapitulate viral entry and the replication of authentic viruses without egress makes them an ideal surrogate with a high degree of safety (20). Most alphavirus VRP packages rely on the coelectroporation of *in vitro*-transcribed (IVT) replicon and helper RNAs into mammalian cells (21). VRPs assembled by transpackaging alphavirus replicons express the gene of interest substituted for structural proteins. Note, however, that alphavirus nsP2 (22) or capsid proteins (23) are key players inhibiting cellular transcription and translation, which can quickly lead to a cytopathic effect (CPE) in mammalian cells, thereby limiting VRP production. Note also that nsP2 and capsid proteins are unable to block cellular events in mosquito cells, which means that they could theoretically be used to achieve alphavirus persistency in a mosquito vector. The use of mosquito cells as an alternative tool for the alphavirus VRP package is worthy of investigation. To the best of our knowledge, no previous study has addressed the packaging of mosquito cell-derived alphavirus VRPs.

Researchers have previously demonstrated the ability of baculoviruses to deliver CHIKV replicons or 26S cDNA into mosquito cell lines (24, 25). Our objective in the current study was to establish an RNA handle-free and recombination-free package system for CHIK VRPs from mosquito cells. This was achieved by constructing a recombinant baculovirus bearing cDNAs of the bipartite two open reading frames (replicon and 26S) of the CHIKV genome under control by either cytomegalovirus (CMV) (replicon RNA expressing dual reporter for enhanced green fluorescent protein [eGFP] and luciferase [Luc]) or hr1pag1 (helper RNA) promoters (Fig. 1A). Our efforts demonstrated that mosquito cell-derived CHIK VRP (mos-CHIK VRP) are easily packaged from mosquito cells via transduction of a single baculovirus vector. We have verified the applicability of mos-CHIK VRP for the rapid quantification of neutralizing (NT) antibodies and antivirals against CHIKV. We have also demonstrated the use of mos-CHIK VRPs in the rapid quantification (6 h) of NT antibodies in serum samples from CHIKV patients and immunized rabbit, thereby indicating their utility in clinical applications. Taken together, the proposed mos-CHIK VRP provides a safe, reliable platform for the rapid quantification of neutralizing antibodies and antivirals.

## RESULTS

**Design and characterization of the recombinant baculovirus.** A mos-CHIK VRP production system (Fig. 1A) was constructed from a recombinant baculovirus bearing a

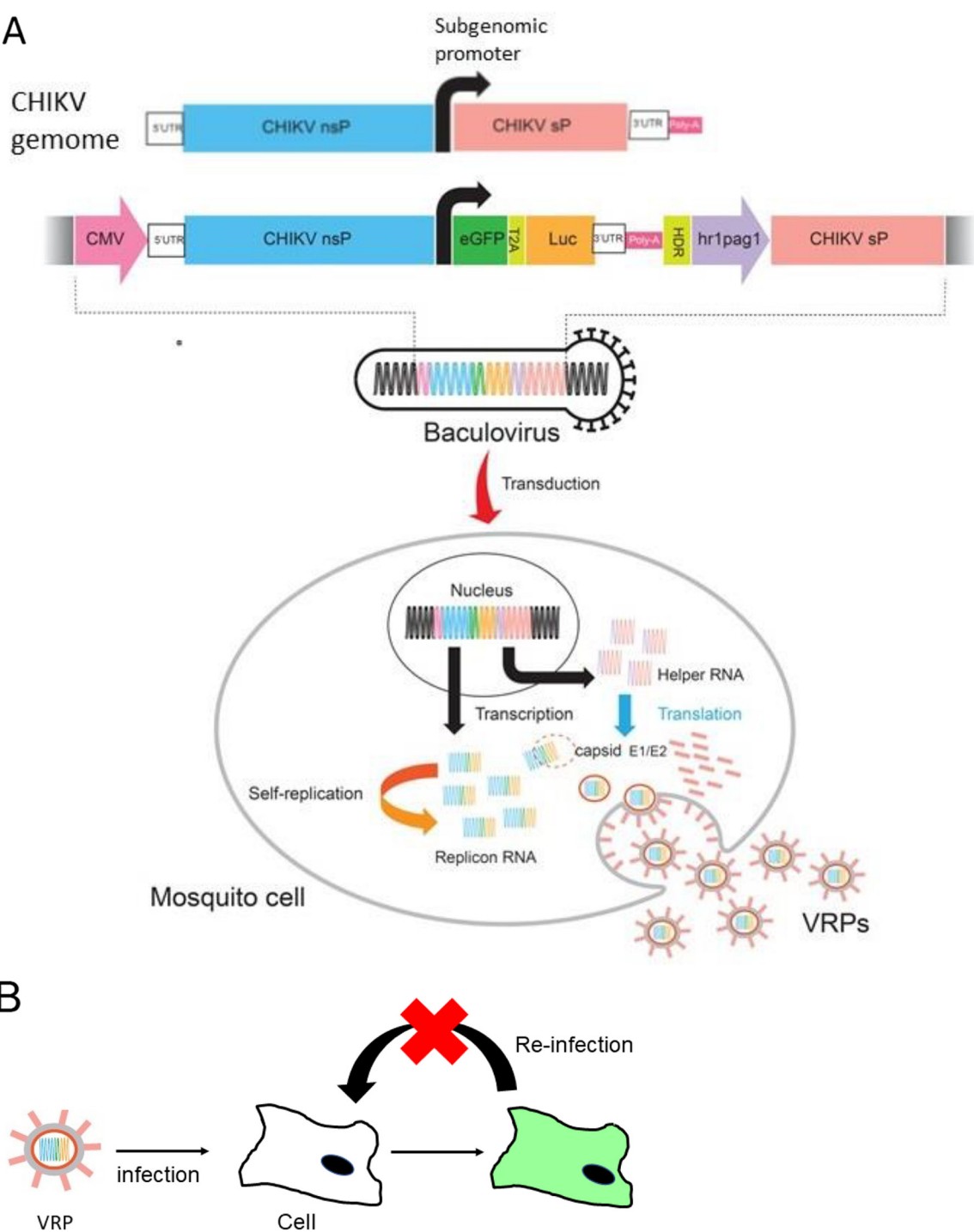

**FIG 1** Schematic illustration showing the Chikungunya virus (CHIKV)-like replicon particle (VRP) production cycle in mosquito cells using a single baculovirus vector. (A) A recombinant baculovirus bearing both DNA cassettes of the CHIKV replicon dual reporter (enhanced green fluorescent protein [eGFP] and luciferase [Luc]) under the control of a cytomegalovirus (CMV) promoter (a shuttle promoter), as well as a helper gene (CHIKV structural protein [sP]) under a hr1pag1 promoter (an insect-specific promoter). Transduction using a single baculovirus vector delivers these genes (the components required for the packaging of VRP) into the nucleus of a mosquito cell for the transcription of self-amplifying replicon RNA and 26S RNA. The VRPs were packaged in a mosquito cell and released. HDR, hepatitis delta virus ribozyme; nsP, nonstructural proteins; T2A, Thosea asigna virus 2A self-cleaving peptides; UTR, untranslated region of CHIKV genome. (B) Single-round infectious VRP infects susceptible cells and expresses reporter genes but does not produce a new VRP for reinfection.

CMV promoter-directed CHIKV replicon containing reporter genes for eGFP and Luc under the transcription control of CHIKV 26S subgenomic promoter and a hr1pag1 promoter-directed 26S gene (CHIKV sP). Western blot analysis was used to detect nsP1, capsid, E2, E1 and eGFP in transduced AP-61 cells in order to characterize the functionality for replicons and helpers (CHIKV sP expression) of CHIKV. Signals specific to nsP1, capsid, E2, E1, and eGFP were detected in transduced AP-61 cells (Fig. 2A). Functionality was confirmed by the fact that transduced AP-61 cells costained with antibodies recognizing nsP1, capsid, E2, eGFP, and/or Luc proteins. Our results revealed the specific colocalization of nsP1, capsid, E2, eGFP, and Luc in transduced AP-61 cells (Fig. 2B), indicating that the recombinant baculovirus expressed all the necessary components.

**Package of mos-CHIK VRP with dual reporter from mosquito cells.** Vero cells were incubated using culture medium from either transduced C6/36 or Ap-61 cells to observe the packaging of CHIK VRPs from transduced mosquito cells. Many eGFP-expressing Vero cells were observed after incubation in control serum for 20 h (Fig. 3A); however, this activity was almost entirely suppressed in serum from CHIKV patients. These results indicate that VRP infection (eGFP expression) was mediated via CHIKV glycoproteins. We sought to confirm mos-CHIK VRP production by performing Western blot analysis on VRP-infected Vero cells using antibodies against nsP1, capsid, and eGFP proteins. Signals specific to nsP1 and eGFP were detected in infected Vero cells; however, we did not observe any signals indicative of the capsid protein (Fig. 3B). VRP production was further confirmed by the fact that infected Vero cells costained with antibodies recognizing nsP1, capsid, eGFP, and/or Luc proteins. Essentially, infected cells presented the specific colocalization of nsP1, eGFP, and Luc without the capsid protein (Fig. 3C). Viral replication in mos-CHIK VRP-infected cells was verified by quantifying the total RNA from VRP-infected cells harvested at either 1 or 24 h postinfection (hpi) via quantitative reverse transcription (qRT)-PCR. As expected, we observed elevated viral RNA levels in treated Vero cells after 24-h incubation (Fig. 3D). Taken together, these results confirmed the packaging of mos-CHIK VRP dual reporter from transduction of mosquito cells with a single baculovirus vector.

**Infection behavior and production kinetics of mos-CHIK VRP.** The effect of temperature on VRP infection was assessed by obtaining VRP titers after incubation for 20 h at 28, 34, or 37°C. The results in Fig. 4A revealed an inverse correlation, as follows: 28°C ($10^4$), 34°C ($10^3$), and 37°C ($10^2$). We also observed a positive correlation between temperature and the severity of VRP-induced CPE. Thus, 28°C was adopted as an optimal temperature for VRP titration. To evaluate the kinetics of VRP production, mosquito cells were transduced using the baculovirus at a multiplicity of infection (MOI) of either 5 or 20. Culture media were harvested at 2- or 3-day intervals and subjected to VRP titration. The results in Fig. 4B show the continuous secretion of mos-CHIK VRPs at concentrations of 1 to $5 \times 10^5$ and 1 to $4 \times 10^4$ VRP/mL from transduced AP-61 and C6/36 cells, respectively. Note that this activity was monitored from 2 to 14 days postinfection (dpi) and peaked at 3 to 5 dpi. Clearly, AP-61 cells outperformed C6/36 cells in terms of mos-CHIK VRP productivity; therefore, AP-61 cells were used for the large-scale preparation of mos-CHIK VRPs. Taken together, our results revealed that mos-CHIK VRP is easily packaged from mosquito cell lines via transduction of a single baculovirus vector.

The effect of temperature on VRP infection was assessed by infecting Vero cells with VRPs at a density of 500 or 2,500 infectious units (IU)/well for incubation at 28, 32, 34, or 37°C for 6 h. The cells were then subjected to Luc assays for the quantitation of VRP infection. The Luc activity results in Fig. 5A rank the temperatures as a function of growth (from high to low) as follows: 34, 32, 37, and 28°C. To confirm the Luc expression kinetics, we subjected VRP-infected Vero cells to Luc assays at intervals of 1 h throughout a total incubation period of 6 h (at 34°C). As shown in Fig. 5B, significant Luc reporter activity began as early as 5 to 6 hpi.

**mos-CHIK VRP is a versatile platform for the rapid quantification of neutralizing antibodies and antiviral screening.** We assessed the effectiveness of mos-CHIK VRP in the rapid screening or quantification of NT antibodies by testing two NT monoclonal antibodies (MAbs) (CHK-265 and 3E7B), four CHIKV patient sera, and one rabbit

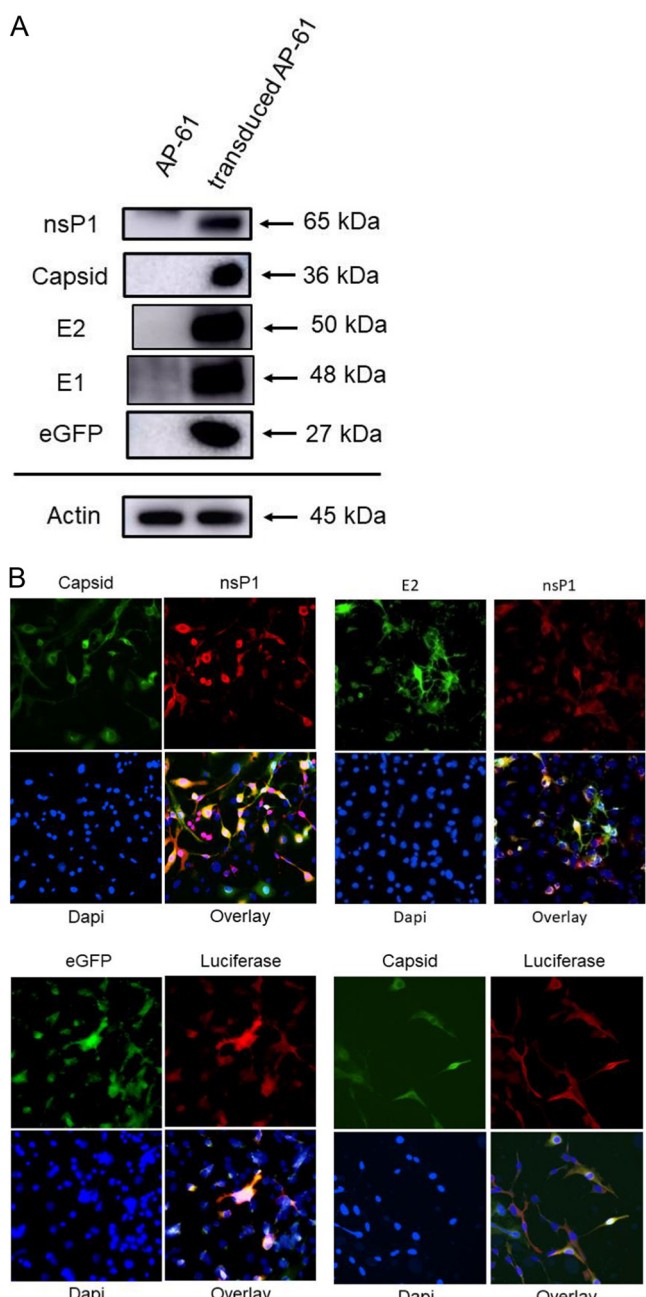

**FIG 2** Characterization of the recombinant baculovirus. (A) Western blot analysis. AP-61 cells were transduced with the baculovirus at a multiplicity of infection (MOI) of 20. Total cell lysates were subjected to Western blot analysis using anti-CHIKV nsP1, capsid, E2, E1, or eGFP (actin served as a loading control) antibodies (shown on the left). The protein sizes (kDa) of markers are indicated by arrows on the right. (B) Colocalization of capsid, E2, nsP1, eGFP, and Luc in transduced mosquito cells. Transduced AP-61 cells were costained using anti-CHIKV capsid/anti-CHIKV nsP1 antibodies (upper left), anti-CHIKV E2/anti-CHIKV nsP1 antibodies (upper right), anti-eGFP/anti-Luc antibodies (lower left), or anti-CHIKV capsid/anti-Luc antibodies (lower right) with 4′,6-diamidino-2-phenylindole (DAPI); overlay figures are presented.

anti-CHIKV serum, all of which were serially diluted and preincubated with mos-CHIK VRP at 37°C for 1 h. The antibody-VRP mixtures were used to infect cells at 34°C for 1 h and then at 34°C for 5 h. We observed a dose-dependent inhibition of VRP infection (Luc) with 50% infective concentration ($IC_{50}$) of 4.2 $\mu$g/mL (CHK-265), 20 $\mu$g/mL (3E7B), 595-fold dilution (patient 1), 4,612-fold dilution (patient 2), 4,239-fold dilution (patient 3), 1,074 -fold dilution (patient 4), and 187-fold dilution (immunized rabbit) (Fig. 6). No

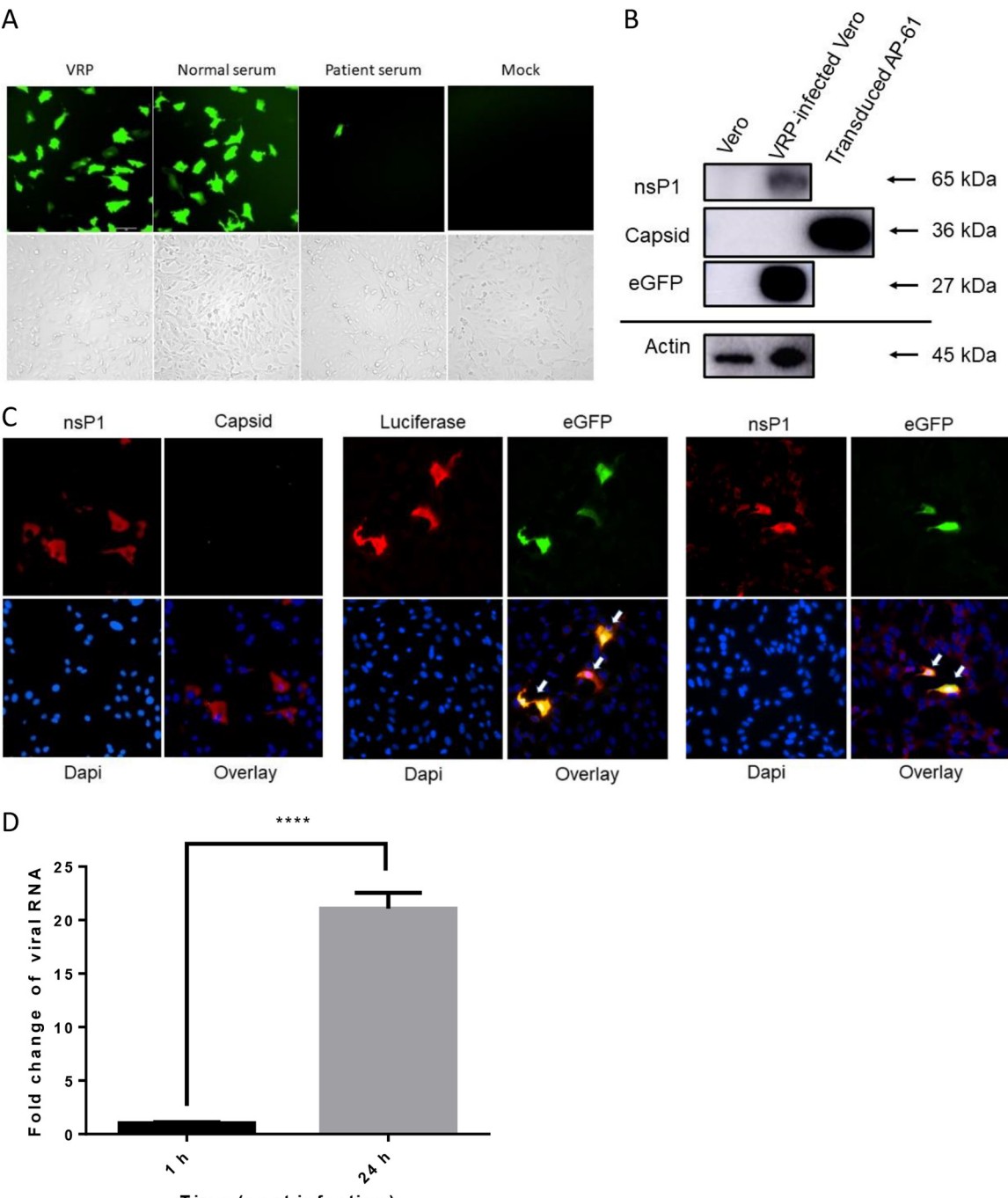

**FIG 3** Package of CHIKV VRPs with dual reporter from mosquito cells. (A) Treating with patient serum and the effects on VRP infection. Vero cells were infected with CHIKV VRPs in the presence of sera of a CHIKV patient or normal individual and incubated for 20 h. The upper panels show the fluorescein isothiocyanate (FITC) channel, and the lower panels show the bright field. (B) Western blot analysis. Vero cells infected with the VRPs at an MOI of 0.05. Total cell lysate was subjected to Western blot analysis using anti-CHIKV nsP1, capsid (included transduced AP-61 cell lysate from Fig. 2A as a positive control), or eGFP (actin served as a loading control) antibodies (shown on the left). Protein sizes (kDa) of markers are indicated by arrows on the right. (C) Colocalization of nsP1, eGFP, and Luc in VRP-infected cells. Panel C shows immunofluorescence imagining of VRP-infected Vero cells: anti-CHIKV capsid/anti-CHIKV nsP1 antibodies (left panels), anti-Luc/anti-eGFP antibodies (middle panels), or anti-CHIKV nsP1/anti-eGFP antibodies (right panels) with DAPI. Merged figures are presented in the lower right panels. White arrows indicate double-positive cells for immunofluorescence assays (IFAs). (D) Viral RNA amplification in VRP-infected cells. Vero cells were infected with VRPs at an MOI of 0.1. Following incubation for 1 or 24 h, the total cellular RNA was harvested and subjected to quantitative reverse transcription (qRT)-PCR. Error bars indicate the standard deviation (SD). Statistical significance was analyzed using Student's $t$ test. ****, $P < 0.0001$.

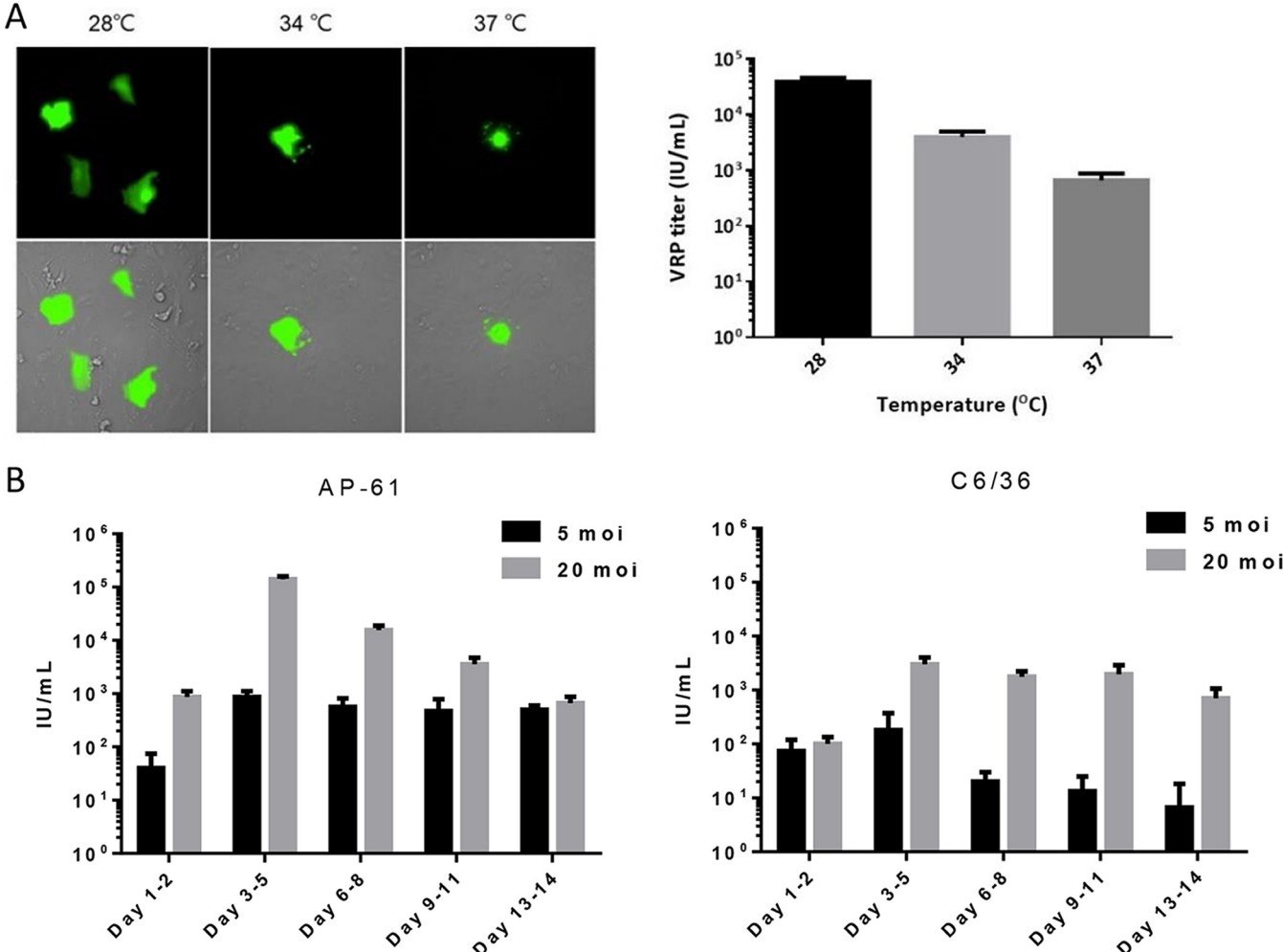

**FIG 4** eGFP expression of CHIK VRP as a function of temperature and production kinetics. (A) VRP titrations obtained at various temperature. The panel shows morphology of eGFP-positive cells incubated at various temperatures (left; 28, 34, and 37°C). eGFP-positive cells at various temperature were counted, and the VRP titer was calculated (right). (B) AP-61 cells or C6/36 cells were transduced using the recombinant baculovirus at MOIs of 5 and 20 in triplicate. Culture supernatant was collected at intervals of 3 days between days 1 and 14 and subjected to VRP titration. Error bars indicate SD.

NT reactivity was detected from treatments with either 6B6C Mab or sera (from a preimmune rabbit and a normal individual). For the sake of comparison, we performed standard CHIKV NT quantification of lenti-CHIKV pseudovirus (17, 18) using the sera from one immunized rabbit and four patients. Pseudovirus-infected cells were analyzed at 72 hpi. As shown in Fig. 6B, we observed little variation in the dose-dependent inhibition of lenti-CHIKV pseudovirus infection, as indicated by the 50% neutralization titer ($NT_{50}$) values pertaining to a 564.7-fold dilution (patient 1), a 1,938.7-fold dilution (patient 2), a 3,450.7-fold dilution (patient 3), a 1,639.1-fold dilution (patient 4), and the 446.2-fold dilution (immunized rabbit) (Fig. 6B). We obtained a Pearson correlation coefficient of $r = 0.8283$ for the $IC_{50}$ values (Fig. 6B). These results demonstrated that mos-CHIK VRP could be used for the rapid quantification of neutralizing activities.

This study also assessed the effectiveness of mos-CHIK VRP as a platform for the testing of anti-CHIKV drugs. This was achieved by assessing the abilities of a viral replication inhibitor (6-azauridine [6-AU] [26]) and a viral entry inhibitor (suramin [27]) in blocking VRP infection (Luc expression). As shown in Fig. 7A, a concentration-dependent inhibition of VRP infection (Luc) was observed within a few hours, as indicated by $IC_{50}$ values of 1.286 µg/mL (6-AU) or 81.3 µM (suramin). These results demonstrated the rapid responsiveness of CHIK VRP (Luc expression) to treatment using 6-AU or suramin. As expected, treatments involving suramin or 6-AU also led to the dose-dependent inhibition of eGFP

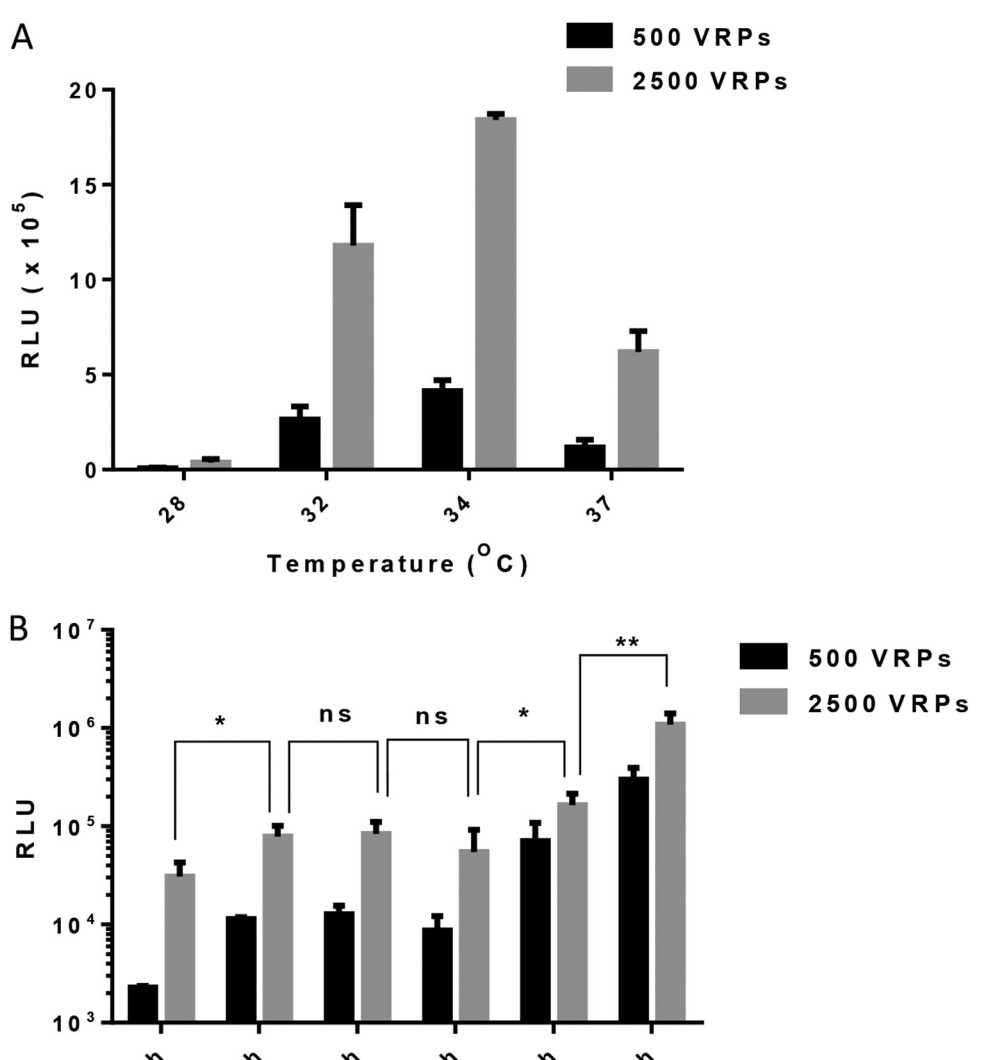

**FIG 5** Luc expression as a function of temperature or expression kinetics. (A) Temperature dependency assay. Vero cells were infected (in triplicate) with VRPs at a density of 500 or 2,500 infectious units (IU)/well and incubated at various temperature for 1 h, whereupon the culture medium was refreshed, and incubation was continued for another 5 h at the same corresponding temperature. The cells were harvested to quantify Luc activity (relative light units [RLU]). (B) Time-course evaluation at 34°C. Vero cells were infected (in triplicate) at a density of 500 or 2,500 IU/well and incubated for 1 h, whereupon the culture medium was refreshed and incubation was continued for another 5 h at 34°C. The cells were harvested at interval of 1 h to quantify Luc activity (RLU). Error bars indicate SD. Statistical significance was analyzed using Student's *t* test. *, $P < 0.05$; **, $P < 0.01$.

expression (another reporter) at 20 hpi (Fig. 7B). Our results indicated a direct correlation between Luc activity and eGFP expression across antiviral treatments. Taken together, these results indicated that mos-CHIK VRP could be used for the rapid quantification and screening for neutralizing antibodies and antivirals targeting CHIKV infection.

## DISCUSSION

Chikungunya fever is a re-emerging public health threat for which antiviral drugs and vaccines are unavailable. As a BSL-3 biological agent, CHIKV research can be performed only in high-level containment facilities. Single-cycle infectious CHIK VRP is antigenically equivalent to live CHIKV (Fig. 1B); however, it is a safe surrogate for CHIKV (i.e., without the risk of reinfection).

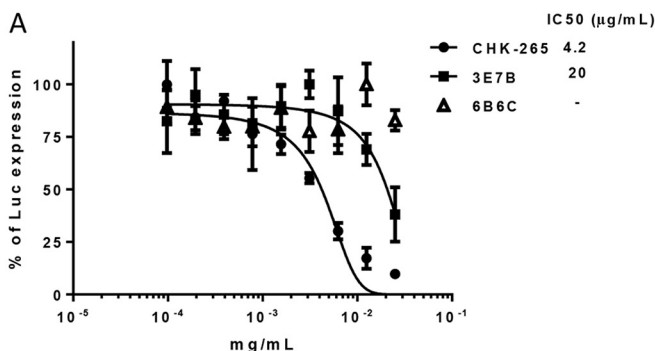

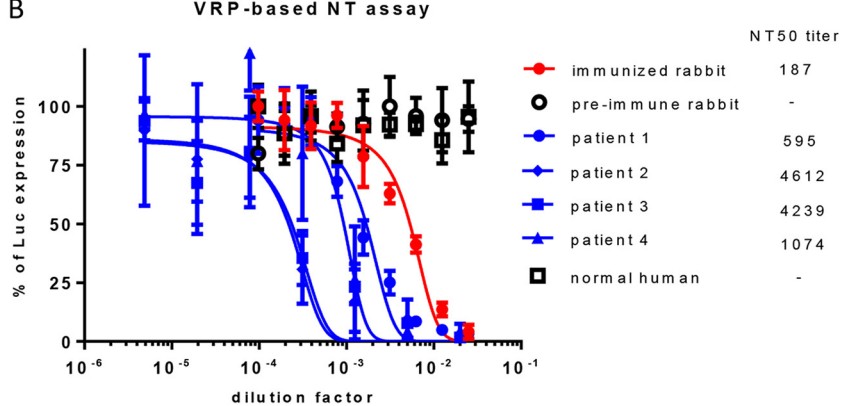

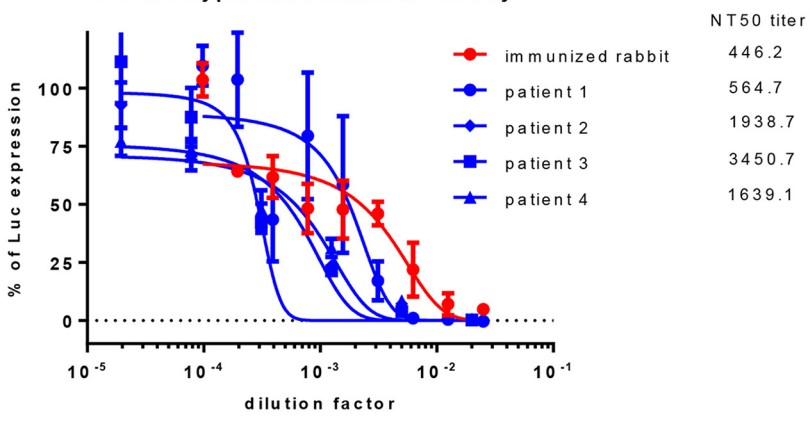

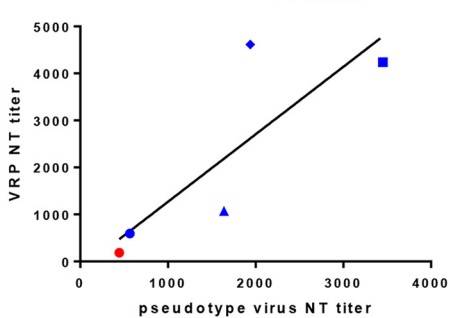

**FIG 6** Rapid measurement of antibody-mediated neutralization using mosquito cell-derived CHIK VRP (mos-CHIK VRP). (A) Monoclonal antibodies (MAbs). mos-CHIK VRP were preincubated with either CHK-265 (a CHIKV-neutralizing Ab), 3E7B (another CHIKV-neutralizing Ab), or 6B6C (negative control) at 37°C for 1 h. The resulting mixtures was then used to infect Vero cell via incubation at 34°C for 1 h. Luc activity was measured after incubation at 34°C for another 5 h and normalized to VRP-infected

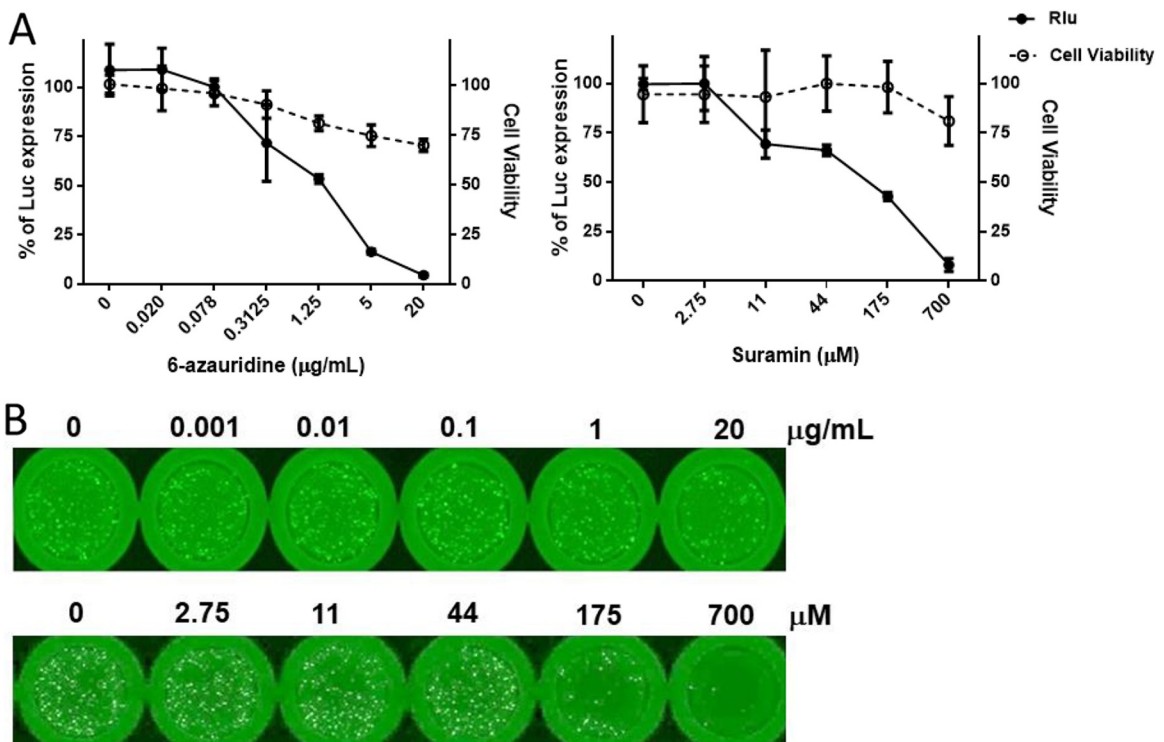

**FIG 7** Rapid quantification and screening of antivirals using mos-CHIK VRP. (A) Rapid antiviral quantification using a Luc reporter. (Left) VRP-infected Vero cells treated with 6-azauridine (6-AU) at indicated concentrations. VRPs were mixed with suramin at different concentration and then infected Vero cells. (Right) The infected Vero cells were treated with suramin at indicated concentration. Luc activity was measured after incubation at 34°C for 5 h and normalized to VRP-infected cell (black lines). Cytotoxicity of 6-AU or suramin on Vero cells (dotted lines) was determined using 3-(4,5-dimethyl-2-thiazolyl)-2,5-diphenyl-2H-tetrazolium bromide (MTT) assay. The data points show the means ± SD of experiments conducted in triplicate, while the error bars indicate the SD. (B) Screening of antivirals using the eGFP reporter. In accordance with panel A, VRP-infected Vero cells were treated with either 6-AU (upper panel) or suramin (lower panel) at the indicated concentrations. The cells were photographed after incubation at 28°C for additional 20 h.

Most alphavirus VRP packages rely on coelectroporation of IVT replicon and helper RNA into mammalian cells (19, 21, 28, 29), thereby limiting VRP packages to cell types with relatively high transfection efficiency. The time and expense of capped IVT-RNA preparation impede VRP utilization. Another major concern is the generation of a replication-competent virus (RCV) via strand switching during replicon and helper RNA replication. The biosafety of RNA-based packaging systems has been greatly enhanced through the use of bipartite helper RNA to provide envelope and capsid genes in *trans* (20, 21, 30) or by knocking out the autoprotease activity of the capsid gene (30). Vaccinia virus-based Venezuelan equine encephalitis (VEE) and Semliki Forest virus (SFV) VRP packaging systems have been shown to obviate RCV by uncoupling the expression of helper RNA with alphavirus replicase/transcriptase in mammalian cells (31, 32).

Our objective in the current study was to generate CHIK VRP from mosquito cells without RCV using genetically modified elements of the CHIKV replicon expressing a dual reporter and 26S gene (helper RNA). Note that this was achieved under the control of CMV and hr1pag1 promoters (24, 25), which were inserted into a recombinant baculovirus vector

**FIG 6** Legend (Continued)

cells. Nonlinear regression analysis was used to calculate the 50% infective concentration ($IC_{50}$) values. Error bars indicate the SD of values averaged from experiments performed in triplicate. (B) Antisera. Samples included the sera from four CHIKV patients and serum from an immunized rabbit. Negative control included serum from a preimmunized rabbit and normal human serum. Nonlinear regression analysis was used to calculate the $NT_{50}$ values. Error bars indicate the SD of values averaged from experiments performed in triplicate. Neutralization titers of sera following with mos-CHIK VRP or CHIKV-pseudotyped virus. The correlation calculated using the Pearson correlation coefficient was $0.7 < r < 0.9$ (strong correlation).

(Fig. 1A). The Bac-Mos vector transiently coexpressed all RNA essential for the packaging of CHIK VRP. Under the control of an insect-specific promoter (hr1pag1), the expression of helper RNA (as mRNA) is independent of CHIKV replicase-transcriptase. Note that removing the alphavirus regulatory signal (subgenomic RNA promoter) from the helper gene separates them from the CHIKV replication compartment, thereby eliminating the possibility of generating crossover recombinants via strand switching. Furthermore, nonreplication baculovirus in mosquito cells (24) could avoid VRP contamination by residual vector, which could generate false-positive helper signals (31, 32). The functionality of recombinant baculovirus was validated through Western blot and the specific colocalization of nsP1, capsid, E2, eGFP, and Luc in transduced mosquito cells (Fig. 2), indicating the expression of functional replicon RNA and helper proteins in mosquito cells.

The efficacy of the CHIK VRP package was assessed by using its culture medium to infect Vero cells. Three lines of evidence (at the RNA, protein, and virus levels) support our assertion that CHIKV VRP was produced from transduced cells. First, abundant viral RNA was detected in infected cells after incubation for 24 h (Fig. 3D). Second, CHIK VRP-infected cells presented the coexpressing of nsP1, eGFP, and Luc, but not the expression of the capsid protein (Fig. 3B and C). Third, treatment using NT antibodies (MAbs and patient sera) (Fig. 3A and 6) or antivirals (suramin and 6-AU) was shown to inhibit CHIK VRP infection (Luc and eGFP expression) (Fig. 7). Kinetic assays indicating CHIK VRP production by transduced mosquito cell lines (AP-61 and C6/36) revealed the continuous secretion of CHIK VRP for a period of 14 dpi (Fig. 4). High CHIK VRP titers were detected in both cell lines over a period of 3 to 5 dpi. The efficiency of CHIK VRP production by AP-61 exceeds that of C6/36.

The optimal temperature for Luc expression in mos-CHIK VRP-infected Vero cells was 34°C (Fig. 5), which is very close to the body surface temperature associated with the mosquito-borne transmission of the arbovirus (33). Note, however, that at 20 hpi, the optimal temperature for eGFP expression in mos-CHIK VRP-infected Vero cell was 28°C (Fig. 4A). The subnormal temperature can perhaps be attributed to the low-temperature environment in which the mosquito host cells were generally cultured. mos-CHIK VRP demonstrated the ability to significantly express the Luc reporter gene within 6 h (Fig. 5B), which agrees with previous reports. Hahn et al. (34) reported on the infection by a recombinant Sindbis virus (another alphavirus), which expressed the chloramphenicol acetyltransferase (CAT) reporter ($>10^6$ molecules/cell) in mammalian cells within 7 h. Hetrick et al. (35) reported on a hybrid alphavirus-severe acute respiratory syndrome coronavirus 2 (SARS-CoV-2) pseudovirus carrying an alphavirus (Semliki Forest virus) replicon utilized for rapid quantification of neutralization antibodies and antiviral drugs against SARS-CoV-2 within 6 h. Gläsker et al. (19) established a Gaussia Luc (Gluc) VRP (mammalian cell-derived) NT assay, which enabled CHIKV NT quantification at 6 hpi. Note, however, that subsequent studies were forced to increase the incubation time to 24 h to measure Gluc activity (36). The incubation time of safety-restricted live CHIKV is 3 to 5 days. The incubation time of pseudotype virus NT assays is 16 to 72 h (18, 37, 38). This incubation time of our NT assay (Luc activity) using mos-CHIK VRP was 6 h at 34°C.

The mos-CHIK VRP-based rapid NT assay (Luc activity) revealed a high correlation coefficient ($r = 0.8283$) with CHIKV envelope protein-pseudotyped virus-based assay (Fig. 6B). In additional, the mos-CHIK VRP-based rapid antiviral assay (Fig. 7) showed the 50% effective concentration ($EC_{50}$) of suramin to inhibit VRP infection (Luc activity) in Vero cells is 81.3 $\mu$M, close to previous report ($EC_{50} = 8.8\sim 62.1$ $\mu$M) based on plaque inhibition assay in various mammalian cell lines (27). The $EC_{50}$ of 6-AU to inhibit VRP infection was 1.28 $\mu$g/mL, which is in good agreement with previous study based on plaque inhibition assay (0.6 to 1.3 $\mu$g/mL) (39). These results were confirmed by the eGFP expression levels of another reporter; however, that approach required a longer incubation period (20 h).

One advantage of our CHIKV VRP dual reporter system is its scalability via baculovirus transduction. Another advantage is the lack of subgenomic RNA promoter in the helper RNA transcript, which obviates the potential emergence of a competent virus

via recombination with replicon RNA (20). Yet another advantage of our CHIKV VRP dual reporter system is its ability to recapitulate entry and replication of the authentic virus without viral budding. Finally, the proposed CHIK VRP system contains both Luc and eGFP reporters. Our results revealed a direct correlation between Luc activity and eGFP expression across treatments involving antibodies or antivirals. That allows for faster (in hours) quantification of NT antibodies and antivirals through Luc activity, which would, however, require expensive Luc reagent, with fewer concerns of the short half-life of antiviral activity or through direct readouts of activities via eGFP signals (20 h). The presence of a dual reporter also allows for the quick elimination of false-positive hits, which would otherwise inhibit Luc or interfere with eGFP fluorescence during high-volume screening (40).

Dendritic cells (DCs) are important initial target cells for arbovirus infection (41). mos-CHIK VRP represents an extremely useful virus surrogate to study how mosquito-borne viruses interact with DCs. Many mosquito cell-derived arboviruses suppress the type I IFN response and enhance the interaction of viral proteins with C-type lectins. This makes many mosquito cell-derived arboviruses more efficient in terms of *in vitro* (on DCs) and *in vivo* infection, compared to viruses derived from mammalian cells (41–44). mos-CHIK VRP perhaps, mimicking the characters of a mosquito cell-derived virus, could be an effective vaccine vehicle compared to mammalian cell-derived VRP.

**Limitations.** The current study was prone to a few limitations, which should be considered in interpreting our findings. First, pseudotyped virus-based neutralization assays are not the gold standard to determine NT titer. More serum samples will be required to compare neutralization titers using VRP versus, for example, the plaque reduction neutralization test.

**Conclusions.** This study demonstrates the facile packaging of mos-CHIK VRP from mosquito cell lines via transduction of a single baculovirus vector. We demonstrated the applicability of the proposed mos-CHIK VRP to the rapid quantification of neutralizing antibodies and antivirals targeting CHIKV. Ease of manufacture, safety, scalability, and high throughput make mos-CHIK VRP a highly valuable vehicle for the study of CHIKV biology, the detection of NT activity, and the screening of antivirals against CHIKV. It also is very likely that the proposed system could be applied to other arboviruses.

## MATERIALS AND METHODS

**Cells.** AP-61 cells (*Aedes pseudoscutellaris*) were cultured in Leibovitz L-15 medium (Gibco) with 10% fetal bovine serum (FBS) (Gibco) and 1% antibiotic-antimycotic (Gibco) at 28°C. C6/36 cells (*Aedes albopictus*) (ATCC CRL-1660) were cultured in RPMI 1640 medium (Gibco) with 10% FBS (Gibco) and 1% penicillin/streptomycin (Gibco) at 28°C under 5% $CO_2$. Vero cells were cultured in minimum essential media (MEM) (Gibco) containing 10% FBS (Gibco), 1% GlutaMAX (Gibco), 1% MEM NEAA (Gibco), 1% sodium pyruvate (Gibco), and 1% penicillin/streptomycin (Gibco) at 37°C under 5% $CO_2$.

**Generation of recombinant baculovirus.** In constructing a two-in-one transfer vector, a modified 4.3-kb synthetic gene of hr1pag1-CHIKV 26S (from the CHIKV OPY-1 strain; GenBank accession number KT449801) was subcloned using PmeI into pFastBac1-VSVG-CHIKV replicon-GFP in tandem with a pair of primers (forward: 5′-TGG CTA AGG GAG GGC GGT TTG TGT TTT ACA AGT AGA ATT CTA CC-3′; reverse: 5′-AGG TAC CGC ATG CTG TTT AAA CTT AGT GCC TGC TGA ACG ACA C-3′) using the NEBuilder HiFi DNA assembly system. The resulting plasmid is here referred to as the pFastBac1-VSVG-CHIKV replicon-GFP-hr1pag1-CHIKV 26S of the two-in-one transfer vector. We then replaced the eGFP gene in pFastBac1-CHIKV replicon-GFP-hr1pag1-CHIKV 26S with a 2.5-kb synthetic gene of eGFP-T2A-Luc using SacI and SpeI sites. The resulting plasmid is here referred to as the pFastBac1-CHIKV replicon-eGFP/Luc-hr1pag1-CHIKV 26S. The generation of recombinant baculoviruses was performed in accordance with the protocol of the Bac to Bac expression system (Invitrogen, Carlsbad, CA, USA). This recombinant baculovirus bearing both DNA cassettes of the CHIKV replicon dual reporter (eGFP and Luc) under the control of a CMV promoter (a shuttle promoter) (24) and a helper gene (CHIKV sP) under a hr1pag1 promoter (an insect-specific promoter) (25). Viral titers were determined using the BacPAK baculovirus rapid titer kit (TaKaRa Bio USA, Inc.) in accordance with the manufacturer's instructions, albeit with the following slight modifications. Fixed cells were permeabilized with 0.5% Triton X-100 in phosphate-buffered saline (PBS) for 5 min prior to immunostaining using anti-CHIKV capsid monoclonal antibodies (Native Antigen Company Ltd.) to replace anti-gp-64 antibodies.

**Western blot analysis.** AP-61 cells were transduced with the recombinant baculovirus at an MOI of 20. After incubation for 24 h, the cells were washed with PBS (Gibco), the culture medium was refreshed, and incubation was continued for another 2 days. Vero cells were infected with CHIKV VRPs at an MOI of 0.05 prior to incubation in MEM at 28°C for 1 h. The cells were then washed using PBS, the culture medium was refreshed, and incubation continued for another 20 h at 28°C. The cells were washed with PBS and lysed in radioimmunoprecipitation assay (RIPA) lysis and extraction buffer (Thermo Fisher Scientific). Protein samples were separated on a 4% to 12% Bis-Tris gel (Invitrogen). Western blot analysis was

conducted following the transfer of proteins onto a nitrocellulose membrane using primary antibodies of either rabbit anti-CHIKV nsp1 polyclonal antibody (GeneTex) (1:5,000), mouse anti-CHIKV capsid MAb (Native Antigen Company Ltd.) (1:1,000), anti-CHIKV E2 MAb (1:1.000) (1b1 from Taiwan CDC), rabbit anti-E1 serum (45), mouse anti-GFP MAb (Santa Cruz Biotechnology, Inc.) (1:1,000), or mouse anti-actin MAb (Sigma-Aldrich) (1:5,000), as well as secondary antibodies of either goat anti-mouse or rabbit secondary antibody conjugated to horseradish peroxidase (Sigma-Aldrich). After incubating the membranes with an ECL substrate, signals were captured using the Amersham Imager 600 (GE Healthcare).

**Immunofluorescence assay.** Immunofluorescence assays (IFAs) of recombinant baculovirus-transduced mosquito cells were obtained by seeding AP-61 cells on chamber slides at $2 \times 10^4$ cells/well. Following incubation overnight, the cells were transduced using the recombinant baculovirus at 0.5 MOI for 24 h. At 3 days post-transduction (dpt), the cells were fixed and costained with rabbit anti-CHIKV nsp1 polyclonal antibody (GeneTex) (1:100), mouse anti-CHIKV capsid MAb (Native Antigen Company Ltd.) (1:100), mouse anti-CHIKV E2 MAb (1:100) (1b1 from Taiwan CDC), mouse anti-GFP MAb (Santa Cruz Biotechnology, Inc.) (1:100), and/or rabbit anti-Luc polyclonal antibody (Abcam) (1:100) and then incubated for 1 h. The cells were subsequently washed three times using PBS and stained with a mixture of Alexa Fluor 594-conjugated and Alexa Fluor 488-conjugated secondary antibodies (Thermo Fisher Scientific) (1:1,000) for 1 h. After the cells were washed using PBS, they were mounted using Fluoroshied mounting medium with 4′,6-diamidino-2-phenylindole (DAPI) (Abcam) for image capture using an inverted fluorescence microscope (DMi8 microscope; Leica Microsystems).

IFAs of CHIKV VRP-infected cells were obtained by seeding Vero cells at $4 \times 10^4$ cells/chamber well. After incubation overnight, the cells were washed using growth medium, infected with CHIKV VRPs, and incubated at 28°C for 1 h before refreshing the medium. At 20 hpi, the cells were subjected to IFA in accordance with the methods described above.

**Serum of CHIKV-infected patient inhibited mos-CHIK VRPs infection (eGFP expression).** Vero cells were seeded at $1.5 \times 10^4$ cells/well (96-well plate). Following incubation overnight, Vero cells were washed using growth medium, infected with VRPs that had been preincubated with 50-fold diluted sera from patient or normal individual at 37°C for 1 h, and then incubated at 28°C for 1 h, whereupon the medium was refreshed and incubation continued at 28°C for another 20 h. These eGFP-expressing cells were photographed using an inverted fluorescence microscope (DMi8 microscope; Leica Microsystems).

**Luc assay.** Luc activity was measured in Vero cells using the Bright-Glo luciferase assay system (Promega) in accordance with the manufacturer's instructions. Measurements were obtained using a SYNERGY H1 microplate reader (BioTek).

**Viral RNA quantification.** Vero cells were seeded at a density of $1 \times 10^5$ cells/well (24-well plate) and incubated overnight, prior to infection via VRPs at an MOI of 0.1. Total cellular RNA was harvested at 1 or 24 h post-transduction (hpt) using a Monarch total RNA miniprep kit (New England Biolabs). One-step quantitative RT-PCR was performed using the One-step RT-PCR kit (Qiagen) in a total volume of 25 $\mu$L. qRT-PCR was conducted using the ABI StepOnePluses real-time PCR system (Applied Biosystems) in triplicate. The primer sequences used in this assay were as follows: Actin-F, 5′-GGC CAG GTC ATC ACC ATT-3′; Actin-R, 5′-ATG TCC ACG TCA CAC TTC ATG-3′ (46); 3′-UTR-F, 5′-CAA ACG GAA GAG ATG TAG G-3′; and 3′-UTR-R, 5′-GGG TTC GGA GAA TCG TGG-3′. Estimates of 3′-UTR expression were based on relative quantification using the comparative critical threshold (CT) method. Viral RNA expression levels were normalized through the endogenous control of actin. Fold changes were calculated using the $\Delta\Delta$CT algorithm based on data obtained at 1 hpt as a reference. The results from two experiments performed independently were very similar; therefore, only the data from a single independent experiment are presented.

**Kinetics of VRP production.** Ap-61 cells ($1 \times 10^5$ cells/well) and C6/36 cells ($3 \times 10^5$ cells/well) were seeded in 12-well plates. After incubation overnight, the cells were transduced in triplicate with the recombinant baculovirus at an MOI of 5 or 20 in growth medium containing 5% FBS. At 24 hpt, the growth medium was refreshed (1.5 mL/well). At 2, 5, 8, 11, and 14 dpt, culture supernatant was harvested, filtered using a 0.22-$\mu$m filter, aliquoted, and stored at −80°C.

**VRP production.** AP-61 cells ($2 \times 10^6$ cells/flask) were seeded in T-25 flasks, incubated overnight, and transduced with recombinant baculovirus at an MOI of 20 in growth medium with 5% FBS. At 24 hpt, the growth medium was refreshed. Live detached cells were observed at 4 dpt. To avoid removing the detached cells, half of the culture supernatant was carefully harvested at intervals of 3 to 4 days to be replaced with fresh growth medium for 12 to 14 dpt. The collected supernatant was centrifuged at 4,000 $\times$ $g$ at 4°C for 20 min, filtered using a 0.22-$\mu$m filter, aliquoted, and stored at −80°C.

**VRP titration.** Vero cells were seeded at $1.5 \times 10^4$ cells/well (96-well plate) 1 day in advance. VRP stock was diluted in a 10-fold serial dilution with MEM, whereupon aliquots of the dilution were added to cultured cells and incubated at 28°C for 1 h, whereupon the medium was refreshed and incubation continued for another 20 h. The number of eGFP-positive cells was counted using an inverted fluorescence microscope (DMi8 microscope; Leica Microsystems).

**Ethics statement.** This study involving clinical human serum samples was reviewed and approved by the Taiwan Centers for Disease Control Institutional Review Board (IRB 109105).

**VRP-based neutralization assay.** Vero cells ($1.5 \times 10^4$ cells/well) were seeded in a 96-well white/clear-bottomed plate (Thermo Scientific Nunc). VRPs were preincubated with serially diluted NT MAbs (CHK-265 [Absolute Biotech] and 3E7B [Novus Biologicals]), as well as sera from patient and inactivated CHIKV-immunized rabbit (45) at 37°C for 1 h. 6B6C MAb (a flavivirus NT antibody [47]), as well as sera from a normal individual and the preimmune rabbit, were used as the negative control. The cells were incubated overnight, washed with growth medium, infected with preincubated VRPs (2,500 IU/well), and incubated at 34°C for 1 h, whereupon the medium was refreshed to continue incubation at 34°C for another 5 h; then the cells were subjected to Luc assay.

**Pseudotyped virus-based neutralization assay.** Synthetic 3.7-kb 26S cDNA from CHIKV strain LR2006_OPY1 (GenBank: KT449801) was subcloned into a pack2 vector to construct pack2-CMV-HHB-26S-poly-A. For packaging CHIKV-26S pseudovirus, $1 \times 10^6$ Lenti-X 293T cells (TaKaRa)/well (6-well plate) were incubated in lentivirus packing medium (Invitrogen) and then transfected with 1.7 $\mu$g pLV-EF1a-Intron-Luc-IRES-Bsd, 1 $\mu$g pack1, and 0.8 $\mu$g pack2-CMV-HHB-26S-poly-A using Lipofectamine 3000 reagent. The cell medium was changed at 6 h post-transfection, and supernatant containing pseudovirus material was harvested and filtered using a 0.45-$\mu$m syringe filter at 24 and 52 h post-transfection. Pseudovirus stock was then stored at $-80$°C. Serum samples were first heat-inactivated over a period of 30 min at 56°C and serially diluted using culture medium containing 1% FBS, before undergoing incubation with $3 \times 10^5$ relative light units (RLU) of the pseudovirus (positive control) or culture medium containing 1% FBS (negative control) at 4°C for 1 h. The mixtures were then dispensed into $1.5 \times 10^4$ Vero cells/well (96-well white/clear-bottomed plate). After incubation at 37°C for 16 h, the culture medium was refreshed (100 $\mu$L/well), whereupon incubation was continued at 37°C for another 48 h cells prior to Luc assay.

**VRP-based antiviral drug assay.** Suramin (Sigma-Aldrich) and 6-azauridine (6-AU) (Sigma-Aldrich) were dissolved in water to produce stock solutions (suramin, 50 mg/mL, 35 mM; 6-AU, 50 mg/mL, 200 mM) and then stored at $-20$°C until use. Vero cells were seeded at a density of $1.5 \times 10^4$ cells/well (96-well white/clear-bottomed plate). For the suramin test, the cells were infected with 2,500 IU VRPs/well in the presence of serially diluted suramin with MEM prior to incubation at 34°C for 1 h. The cells were then grown for another 5 h in 5% FBS medium containing corresponding concentrations of suramin at 34°C. The 6-AU test involved infecting cells with 2,500 IU VRP/well followed by incubation at 34°C for 1 h. The cells were then grown for another 5 h in 5% FBS medium containing corresponding concentrations of 6-AU at 34°C and then subjected to Luc assay. eGFP (another reporter) expression was assessed by incubating treated cells at 28°C for 20 h. Images were then captured using a Sapphire biomolecular imager (Azure Biosystems).

**3-(4,5-Dimethyl-2-thiazolyl)-2,5-diphenyl-2H-tetrazolium bromide (MTT) assay.** Cell viability was determined using the Cell Counting kit-8 (Enzo) according to the manufacturer's instructions. Absorbance at 450 nm was measured using a microplate reader Infinite 200 PRO (Tecan Trading, Ltd., Switzerland). Cell viability was evaluated in terms of absorbance at various doses divided by the absorbance at 0 $\mu$g/mL.

**Statistical analysis.** The data were analyzed using GraphPad Prism 6.01 software and assessed for statistically significant differences using a two-tailed, unpaired $t$ test. A $P$ value $< 0.05$ was considered statistically significant; $P$ values were indicated as follows: ns, not significant; *, $P < 0.05$, significant; **, $P < 0.01$, highly significant; ***, $P < 0.001$ and ****, $P < 0.0001$, extremely significant. The half-maximal inhibitory concentration (or dilution) was calculated using the nonlinear repression analysis. The interrelationship between the mos-CHIK VRP-based NT titer and CHIKV-pseudotyped virus-based NT titer was calculated via correlation analysis.

## ACKNOWLEDGMENTS

We declare no conflict of interest.

Financial support was received from the NHRI (NHRI-110A1-MRCO-08212101 and NHRI-11A1-MRCO-08222201 to S.-C.K.), MOST (MOST110-2740-B-016-001 and NSCT 111-2327-B-016-002 to S.-C.K.), Medical Affairs Bureau (MND-MAB-110-075 and MND-MAB-D-111081 to S.-C.K.), and National Defense Medical Center (IPM-110-G7 and 111-P-12 to S.-C.K.) (IPM-110-G7-4 and 111-G4-2 to H.-C.L) of Taiwan.

H.-C.L.: Data curation, Formal analysis, Funding acquisition, Investigation, Methodology, Validation, Visualization, Writing; D.-J.C.: Conceptualization, Investigation, Methodology; P.-Y.S.: Methodology; H.-T.L.: Investigation; C.-C.H.: Investigation; C.-C.L.: Methodology; S.-C.K.: Conceptualization, Data curation, Funding acquisition, Methodology, Project administration, Supervision, Validation, Writing. All authors have read and agreed to the published version of the manuscript.

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
