## [Reviewer comments · Microbiology Spectrum]

Microbiology Spectrum

Development of a novel chikungunya virus-like replicon particle for rapid quantification and screening of neutralizing antibodies and antivirals

Szu-Cheng Kuo, Hui-Chung Lin, Der-Jiang Chiao, Pei-Yun Shu, Hui-Tsu Lin, Chia-Chu Hsiung, and Chang-Chi Lin

Corresponding Author(s): Szu-Cheng Kuo, National Defense Medical Center

Review Timeline:

Submission Date:	November 24, 2022
Editorial Decision:	January 27, 2023
Revision Received:	February 8, 2023
Accepted:	February 9, 2023

Editor: Juan Ludert

Reviewer(s): The reviewers have opted to remain anonymous.

Transaction Report:

DOI: <https://doi.org/10.1128/spectrum.04854-22>

January 27, 2023

Dr. Szu-Cheng Kuo
National Defense Medical Center
P.O. BOX 90048-700, San Hsia Dist.
New Taipei City
Taiwan

Re: Spectrum04854-22 (Development of a novel chikungunya virus-like replicon particle for rapid quantification and screening of neutralizing antibodies and antivirals)

Dear Dr. Szu-Cheng Kuo:

Thank you for submitting your manuscript to Microbiology Spectrum. Your manuscript was revised by an expert in the field who raised several concerns I would kindly ask you to address. In addition, after reading the paper myself, I also have a few suggestions and comments: 1) Add to Figure 2 some experimental evidence of the expression of E1 and E2 proteins from the baculovirus construction; 2) Add to the Material and Methods section a brief explanation for the rationale of using promoters of 2 different origins (i. e., vertebrate and mosquito promoters); and 3) Please consider shortening the Discussion section by avoiding repetitions with the Result section.

When submitting the revised version of your paper, please provide (1) point-by-point responses to the issues raised by the reviewer and myself as file type "Response to Reviewers," not in your cover letter, and (2) a PDF file that indicates the changes from the original submission (by highlighting or underlining the changes) as file type "Marked Up Manuscript - For Review Only". Please use this link to submit your revised manuscript - we strongly recommend that you submit your paper within the next 60 days or reach out to me. Detailed instructions on submitting your revised paper are below.

Link Not Available

Sincerely,

Juan E. Ludert

Journals Department
Reviewer comments:

Reviewer #1 (Comments for the Author):

In this study by Lin et al., the authors describe the development of a novel virus-like replicon particle (VRP) system to study the chikungunya virus (CHIK-VRP) under BSL2 conditions. In this VRP system, the structural protein ORFs under the control of viral subgenomic promoter were replaced with an eGFP-luciferase dual reporter. Single-cycle infectious VRP were produced in mosquito cells by using a baculovirus vector that co-expresses both the viral replicon and structural proteins. The authors

perform several validation studies for their VRP system and report that the VRPs recapitulate authentic CHIK life cycle in several assays including in vitro neutralization with human patient serum, inhibition of replication with small molecule inhibitors, etc. Finally, in the viral neutralization assays, this VRP system showed higher sensitivity and reproducibility than lentivirus-based CHIKV pseudoparticles. These results demonstrate that this newly developed mos-VRP-CHIKV can serve as a better tool to study CHIKV under BSL2 conditions. Overall, this manuscript describes the development of a novel tool that might be of interest to alphavirus researchers. The authors should address the following concerns:

1. Figure 1A is a little confusing. Please include the schematics for WT CHIKV genome organization alongside with the VRP; also clearly indicate 5'UTR, 3'UTR, and sgRNA promoters in the schematics.
2. The authors should assess the stability of GFP-Luc reporter genes. In Fig 3B-c: very few cells are expressing GFP as compared to nsp1, which raises concerns about the stability of dual reporter transgene in this VRP system.
3. In Figure 4: The authors only show the viral titers in their assessment of continuous VRP production. Do the mosquito cell lines remain viable during the entire 14-day testing period? How can you differentiate between initially produced VRP remaining stable in the supernatant vs newly produced?
4. Panel numbering is confusing for some figures as the authors use A, B, C for main panels and a,b and c for sub-panels. This needs to be rectified.
5. LN 191-194: Fig S2 should be referred in the text.
6. LN200: Figure number is incorrect. It should be 6C not 5C.
7. Some of figures could be combined to reduce the total number of figures. Supplementary figures could be merged into main figures.

Staff Comments:

Preparing Revision Guidelines

Please return the manuscript within 60 days; if you cannot complete the modification within this time period, please contact me. If you do not wish to modify the manuscript and prefer to submit it to another journal, please notify me of your decision immediately so that the manuscript may be formally withdrawn from consideration by Microbiology Spectrum.

editor

- 1) Add to Figure 2 some experimental evidence of the expression of E1 and E2 proteins from the baculovirus construction;
Response: We thank the reviewer for the precious suggestion. Revised Figure 2 has included evidence (WB and IFA) of the expression of E1 and E2 proteins. The Text has also been revised at LN 127,129,130, 253, 391-2, 405 and 735-51.
- 2) Add to the Material and Methods section a brief explanation for the rationale of using promoters of 2 different origins (i. e., vertebrate and mosquito promoters);
Response: We thank the reviewer for the good recommendation. Material and Methods section has been revised to include a brief explanation and references for the rationale of using promoters of 2 different origins at LN 368-371.CA, USA). **This recombinant baculovirus bearing both DNA cassettes of the CHIKV replicon-dual reporter (eGFP and Luc) under the control of a CMV promoter (a shuttle promoter) (24) and a helper gene (CHIKV sP) under a hr1pag1 promoter (an insect cell-specific promoter) (25).** Viral titers were....
- 3) Please consider shortening the Discussion section by avoiding repetitions with the Result section.
Response: We thank reviewer for the valuable suggestion. Discussion section has been shortening to avoid repetitions with the Result section.

Reviewer

1. Figure 1A is a little confusing. Please include the schematics for WT CHIKV genome organization alongside with the VRP; also clearly indicate 5'UTR, 3'UTR, and sgRNA promoters in the schematics.
Response: Thanks for your good suggestion. WT CHIKV genome organization alongside with the VRP; also clearly indicates 5'UTR, 3'UTR, and sgRNA promoters have been included in the schematics of revised Figure 1A.
2. The authors should assess the stability of GFP-Luc reporter genes. In Fig 3B-c: very few cells are expressing GFP as compared to nsp1, which raises concerns about the stability of dual reporter transgene in this VRP system.
Response: We thank the reviewer for pointing out the non-specific issue. The first antibody (rabbit anti-CHIKV nsp1 serum) for CHIKV nsp1 staining is a polyclonal antibody with background signal in IFA. Therefore, false positive cells were observed in nsp1 staining. To avoid non-specific interference, revised Fig 3B has included white arrows to specifically indicate double-positive cells for IFA.
3. In Figure 4: The authors only show the viral titers in their assessment of continuous VRP production. Do the mosquito cell lines remain viable during the entire 14-day testing period? How can you differentiate between initially

produced VRP remaining stable in the supernatant vs newly produced?

Response: We thank the reviewer for important issues.

Viabilities of both transduced mosquito cells (C6/36 and AP-61) at an MOI of 20 were declined at 5 dpt. As much lower viability of transduced C6/36 cells comparing the viability of transduced AP-61 cells was observed. Rare mosquito cells keep attachment on flask at 8 dpt. However, comparable VRP titers were harvested during dpt 9-14.

We recently found equal VRP infectivity on both Vero and AP-61 cells (data not shown), which indicates that initially produced VRP could infect mosquito cells. Thus, the initially produced VRP could be unstable in the supernatant vs the newly produced VRP. Shortening interval time to prevent VRPs infect mosquito cells could improve VRP yield.

4. Panel numbering is confusing for some figures as the authors use A, B, C for main panels and a, b and c for sub-panels. This needs to be rectified.

Response: We thank the reviewer for the valuable suggestion. Labels (a, b and c) for sub-panels of Figure 2 and 3 have been omitted.

5. LN 191-194: Fig S2 should be referred in the text.

Response: We thank reviewer for the good suggestion. Fig S2 has been merged into Fig 6.

6. LN200: Figure number is incorrect. It should be 6C not 5C.

Response: It has been revised on LN202.

7. Some of figures could be combined to reduce the total number of figures.

Supplementary figures could be merged into main figures.

Response: We thank the reviewer for the good suggestion. Labels of sub-panels of Fig 2B and 3C have been omitted to reduce the total number of figures. Some figures of Fig 6 have been combined to reduce the total number of figures.

Supplementary figures have been merged into Fig 4 and 6.

February 9, 2023

Dr. Szu-Cheng Kuo
National Defense Medical Center
P.O. BOX 90048-700, San Hsia Dist.
New Taipei City
Taiwan

Re: Spectrum04854-22R1 (Development of a novel chikungunya virus-like replicon particle for rapid quantification and screening of neutralizing antibodies and antivirals)

Dear Dr. Szu-Cheng Kuo:

I am glad to inform you that your manuscript has been accepted, and I am forwarding it to the ASM Journals Department for publication. You will be notified when your proofs are ready to be viewed.

Publication Fees: We have partnered with Copyright Clearance Center to collect author charges. You will soon receive a message from no-reply@copyright.com with further instructions. For questions related to paying charges through RightsLink, please contact Copyright Clearance Center by email at ASM_Support@copyright.com or toll-free at +1.877.622.5543. Hours of operation: 24 hours per day, 7 days per week. Copyright Clearance Center makes every attempt to respond to all emails within 24 hours. For a complete list of **Publication Fees**, including supplemental material costs, please visit our website.

Sincerely,

Juan E. Ludert
Editor, Microbiology Spectrum
